# Gaze Estimation Approach Using Deep Differential Residual Network

**DOI:** 10.3390/s22145462

**Published:** 2022-07-21

**Authors:** Longzhao Huang, Yujie Li, Xu Wang, Haoyu Wang, Ahmed Bouridane, Ahmad Chaddad

**Affiliations:** 1School of Artificial Intelligence, Guilin University of Electronic Technology, Jinji Road, Guilin 541004, China; longzhaohuang1722@gmail.com (L.H.); 1901610215@mails.guet.edu.cn (X.W.); 1901620216@mails.guet.edu.cn (H.W.); 2Faculty of Engineering and Environment, Northumbria University, Newcastle NE18ST, UK; abouridane@sharjah.ac.ae; 3The Laboratory for Imagery Vision and Artificial Intelligence, Ecole de Technologie Superieure, 1100 Rue Notre Dame O, Montreal, QC H3C1K3, Canada

**Keywords:** gaze estimation, gaze calibration, noise image, differential residual network

## Abstract

Gaze estimation, which is a method to determine where a person is looking at given the person’s full face, is a valuable clue for understanding human intention. Similarly to other domains of computer vision, deep learning (DL) methods have gained recognition in the gaze estimation domain. However, there are still gaze calibration problems in the gaze estimation domain, thus preventing existing methods from further improving the performances. An effective solution is to directly predict the difference information of two human eyes, such as the differential network (Diff-Nn). However, this solution results in a loss of accuracy when using only one inference image. We propose a differential residual model (DRNet) combined with a new loss function to make use of the difference information of two eye images. We treat the difference information as auxiliary information. We assess the proposed model (DRNet) mainly using two public datasets (1) MpiiGaze and (2) Eyediap. Considering only the eye features, DRNet outperforms the state-of-the-art gaze estimation methods with *angular-error* of 4.57 and 6.14 using MpiiGaze and Eyediap datasets, respectively. Furthermore, the experimental results also demonstrate that DRNet is extremely robust to noise images.

## 1. Introduction

Eye gaze is an important nonverbal communication technology. It contains rich information about human features, allowing researchers and users to tap more about human patterns [1,2] and action [3,4]. It is widely recommended in many topics, e.g., human–robot interaction (HRI) [5,6,7,8]. Most common gaze estimation tasks are categorized into three types: (1) three-dimensional (3D)-based gaze estimation [9], (2) target estimation [10,11] and (3) tracking estimation [12]. Figure 1 shows examples of gaze estimation task types. However, our study focuses on 3D gaze estimation.

Three-dimensional gaze estimation can be classified into two methods, as illustrated in (Figure 2). Model-based methods [13,14,15,16] generally consider geometric features such as eyeball shape, pupil center position, and pupil membrane edge. These methods require specific equipment such as infrared camera and have low robustness when illumination and head pose change. However, appearance-based methods have higher performance due to the training of a deep network using a large amount of data. Specifically, the deep network has the ability to extract features from eye images under various illumination conditions and head positions. Only a laptop with a web camera is required to collect the data set (e.g., MpiiGaze [17]).

The first appearance-based method [18] uses convolutional neural networks (CNN) inherited from LeNet [19] for gaze estimation. One factor that limits CNN success is the noise in the eye images. Figure 3 shows the noise images caused by the extreme head position and the blink response in Eyediap [20]. The left eye image in (a) is not completely captured due to the extreme head position. The left and right eye images missed the pupil information in (b) due to the blink response. We aim to avoid the limitation of the noisy eye images using the proposed DRNet model (i.e., more details are given in Section 4.2).

Gaze calibration problem also limits the performances of CNNs. An effective and simple solution proposed by many publications to solve this problem is to adjust the weight of the model after training [11,21,22,23,24]. However, this solution requires many inference images with the label. Liu et al. [25] propose a differential network (Diff-Nn) to address the gaze calibration problem by directly predicting the difference information between two images of the eyes. Gu et al. [26] developed Diff-Nn for the gaze estimation using the left and right eye patch of one face simultaneously. Several other works mention that the performance based on the methods considering the difference information is directly affected by the number and the specific label of the inference image [25,26].

We firstly treated the difference information as auxiliary information in the proposed DRNet. We combined the original gaze direction and the difference information through the shortcut-connection in DRNet. In addition, we proposed a new loss function for the gaze estimation. For example, the original loss function evaluates the gap between the quantity of the predicted vector and its ground truth, such as pitch and yaw. The new loss function evaluates the intersection angle between the predicted and its ground truth vector in 3D space directly.

To the best of our knowledge, this is the first study that applies the shortcut-connection by combining the difference information to address gaze calibration. Our contributions can be summarized as follows.

We propose the DRNet model, which applies the shortcut connection, to address the gaze calibration problem and hence improve the robustness-to-noise image in the eye images. DRNet outperforms the state-of-the-art gaze estimation methods only using eye features, and is also highly competitive among the gaze estimation methods combining facial feature.We propose a new loss function for gaze estimation. It provides a certain boost to existing appearance-based methods.

The remainder of this paper is structured as follows. The related works are presented in Section 2. Section 3 describes the proposed pipeline-based DRnet. We present the experimental results in Section 4. Finally, Section 5 concludes the key contributions of our work.

## 2. Related Work

In previous years, appearance-based methods have been considered as the most commonly methods in gaze estimation. For example, Zhang et al. [18] proposed the first appearance-base method (i.e., LeNet [19]) that uses eye features for gaze estimation. They expanded three convolution layers to sixteen convolution layers in their work [17] to achieve higher performance metrics. Fischer et al. [27] presented a two-stream network; left and right eye images are fed into VGG-16 [28] separately. Some studies directly used face images as input or/and applied CNN to automatically extract deep facial features. For example, Zhang et al. [29] used a spatial weighting mechanism to efficiently encode the face location using CNN. This method decreases noise impact and improved the contribution of highly activated regions. Cheng et al. [30] assigned weights for two eye features under the guidance of facial features. Furthermore, Chen et al. [31] considered dilated convolution to extract deep facial features. This effectively improves the perceptual field while reducing the image resolution [31]. In addition, gaze estimation in outdoor environments was investigated using eye and face features derived from near-infrared camera [32,33]. Bao et al. [34] studied a self-attention mechanism to combine two eye features with the guidance of facial features. In [35], CNN with long short-term memory (LSTM) network is introduced to be able to capture spatial and temporal features from video frames. In [36], the generative adversarial network is used to enhance the eye image captured under low and dark light conditions. Despite all the advantages of gaze estimation techniques, there are still some challenges that need to be addressed.

In order to avoid the challenges of previous gaze estimation techniques, we developed DRNet to treat the difference information as auxiliary information and designed the model based on the residual concept. It is worth noting that the residual network concept was first proposed by He et al. [37] to avoid the model degradation problem of deep neural networks. For example, in residual networks, increasing the depth of the network does not result in decreasing the accuracy due to the shortcut connection. Thus, we apply the shortcut connection in DRNet to improve the robustness of the differential network.

## 3. Methodology

This paper proposes a DRNet model with a new loss function to optimize the performance of gaze estimation. Specifically, the difference information is used as an auxiliary information in DRNet model. A brief overview of the DRNet model with the proposed loss function are detailed as follows.

### 3.1. Proposed DRNet

Figure 4 shows the proposed DRNet pipeline. It consists of a feature extractor, differential (DIFF), adjustment (AD), and shortcut (SC) modules. Specifically, DRnet receives two eye images (i.e., test and guidance images), and one of these eye images (i.e., guidance image) represents the calibration image. Furthermore, two eye input images are required to be derived from the same person.

#### 3.1.1. Feature Extractor

Instead of one single eye image, both test and guidance eye images are adopted as raw input for DRNet. The feature extractor is stacked by the convolution layer (Conv), the batch normalization layer (BN) and the rectified linear unit (ReLU). The features are then used as derived from the fully connected layers.

#### 3.1.2. Residual Branch

The three other components (i.e., DIFF, AD, and SC modules) construct the residual branch of the proposed DRNet architecture. More specifically, the DIFF module is responsible for providing the difference information between the test and guidance images. The AD module converts the difference information to the auxiliary information. The SC module provides the gaze-estimation-based information of the test image. Finally, the gaze direction represents the summation of SC and AD outputs.

### 3.2. The Residual Structure in DRNet

Figure 5 shows a block diagram of residual structure process. Guidance and test image features are extracted by the feature extractor. These features are the input of the DIFF Module. In addition, test image feature is transferred into SC Module separately.

It is worth noting that the residual structure of our DRNet model is designed based on the ResNet architecture [37]. Referring to the idea of a shortcut connection in ResNet [37], DRNet combined the difference information and gaze direction through the shortcut connection. The residual structure in DRNet is constructed by the fully-connected layer, while the residual structure in the ResNet is constructed based on the convolutional layers. Therefore, the residual structure of ResNet is an operation on the feature map, and the final output is the sum of two feature maps. Thus, the residual structure of DRNet operates on a one-dimensional vector, while the final output is the sum of two one-dimensional vectors.

### 3.3. Loss Function

We propose a new and original loss function as follows:(1)$$L_{\mathrm{new}}=\frac{|g_{DRNet}|\left|{\hat{g}}_{test}\right|}{\sqrt{g_{DRNet}}\sqrt{{\hat{g}}_{test}}},$$
(2)$$L_{\mathrm{original}}=\left|g_{DRNet}-{\hat{g}}_{test}\right|,$$
where $g_{DRNet}$ is the DRNet output, $g_{test}$ is the test image (e.g., the ground truth).

The loss functions $L_{\mathrm{new}}$ and $L_{\mathrm{original}}$ measure the angle and difference between the predicted vector and the ground truth vector, respectively. It is noted that the $L_{\mathrm{new}}$ loss function uses approximate global information to optimize the output. Another loss function (named LB) based on a combined loss function $L_{\mathrm{new}}$ and $L_{\mathrm{original}}$. LB can be expressed as follows:(3)$$LB\;=\;\alpha *\frac{|g_{DRNet}|\left|{\hat{g}}_{test}\right|}{\sqrt{g_{DRNet}}\sqrt{{\hat{g}}_{test}}}+\left(1-\alpha \right)*\left|g_{DRNet}-{\hat{g}}_{test}\right|,$$
where α is the hyperparameter tuning $L_{\mathrm{new}}$ and $L_{\mathrm{original}}$.

We note that LB plays the role of optimization in DRNet.We also optimized the DIFF module using the following loss function LA.
(4)$$LA=\left|\frac{\left|g_{diff}\right|\left|{\hat{g}}_{guidance}\right|}{\sqrt{g_{diff}}\sqrt{{\hat{g}}_{guidance}}}-\frac{\left|{\hat{g}}_{test}\right|\left|{\hat{g}}_{guidance}\right|}{\sqrt{{\hat{g}}_{test}}\sqrt{{\hat{g}}_{guidance}}}\right|,$$
where $g_{diff}$ and ${\hat{g}}_{guidance}$ is the DIFF output and guidance image (i.e., ground-truth), respectively. We use the loss function LA to measure the difference information of the DIFF module.

Compared to the loss function described in Diff−Nn [25], LA also optimizes prediction by measuring difference information. In other words, LA tunes the prediction to a reasonable scale. The process advantage of LA is that the guidance image label will not be involved in the testing stage, while in Diff−Nn some label information is needed.

The general loss function *L* combined LA and LB as follows:(5)L=(1−β)∗LA+β∗LB,
where β is a hyperparameter tuning of LA and LB.

### 3.4. Training Model

Figure 6 shows the pipeline of the training model. (1) Initialization: Test image is fixed and guidance image randomly selected. (2) Forward propagation: Calculate the output of each unit, and the deviation between the target value and the actual output. (3) Backward propagation: compute the gradient and update the weight parameters. When the iteration reaches the maximum epoch, the DRNet parameters considered and fixed for the prediction. We implement DRNet using PyTorch (https://pytorch.org/, accessed on 2 February 2020) that runs on TITAN RTX GPUs. We considered Adam optimizer with an initial learning rate of 0.01 (decayed by 0.1 every 5 epochs) and batch size of 128 and 1 in training and testing, respectively.

## 4. Experiments

To validate our proposed architecture, two public datasets have been used in the experimentation process: (i) MpiiGaze dataset and (ii) Eyediap dataset. An example of sample eye images in Eyediap and MpiiGaze is shown in Figure 7.

(1) MpiiGaze dataset consists of 1500 left and right eye images derived from 15 subjects [17]. These images are obtained in real-life scenarios with a variation of the illumination conditions, head pose and subjects with glasses. Specifically, the images are grayscale with a resolution of 36×60 pixels with corresponding information related to head pose.

(2) Eyediap dataset consists of 94 videos taken from 16 subjects [20]. The data set is obtained in a laboratory setting with the corresponding head pose and gaze. Data sets were pre-processed following the pre-processing procedures described in [38], and cropping approximately 21K images of the eyes, which are also grayscale images with a size of 36 × 60 pixels. Note that since two subjects lack the videos in the screen target session, we obtained the images of 14 subjects in our experiments.

We used *angular-error* as a measurement which is generally used to measure the accuracy of the 3D gaze target method as follows:(6)$$angle\_error=\frac{\left|g_{test}\right|\left|{\hat{g}}_{test}\right|}{\sqrt{g_{test}}\sqrt{{\hat{g}}_{test}}}.$$
where ${\hat{g}}_{test}$ and $g_{test}$ is the true and predicted test image for gaze direction, respectively.

### 4.1. Appearance-Base Methods

Table 1 reports the *angular-error* of the appearance-based methods using eye features. Compared to baseline methods (Mnist [18], GazeNet [17], RT-Gen [27], DenseNet101-Diff-Nn [25]), our proposed DenseNet101-DRNet delivers the best performance using the features of the eye image of two public datasets (Figure 7). It is worth noting that the loss function used in DRNet is based on Equation (Equation 5), where α = β = 0.75.

We also compared the performance of the proposed DRNet with the baseline methods using eye and facial features (i.e., Dilated-Net [31], Full Face [29], Gaze360 [39], AFF-Net [34], CA-Net [30]). We note that DRNet model uses only eye features. In related works, the performance of methods using eye and facial features show higher performance than the methods using only the eye features. Table 2 reports the performance of *angular-error*. It can be seen that the proposed DenseNet101-DRNet is highly competitive among the gaze estimation methods (Figure 8). DRNet model is better than Dilated-Net [31], FullFace [29], AFF-Net [39] using Eyediap dataset. In addition, DRNet shows a better performance than FullFace [29] using MpiiGaze dataset.

In addition, we assess DenseNet101-DRNet using the Columbia gaze dataset (CAVE-DB) [40]. We found that the DenseNet101-DRNet using CAVE-DB shows the lowest *angular-error* of 3.70 compared to 4.57 and 6.14 using MpiiGaze and Eyediap datasets, respectively.

### 4.2. Noise Impact on DRNet Model

To study the impact of noise impact on the proposed DRNet architecture, we have adopted RT-Gene (RT-Gene [27] is a model using two eye images where the left and right eye patches are fed separately to VGG-16 networks [28] allowing us to perform feature extraction) as a two-stream model. This scenario is used with an input using two images of the eye [27]. Figure 9 shows an example of a two-stream model. It consists of a feature extractor (e.g., convolution layers) and a regression (e.g., fully connected layers) modules. Two eyes images (i.e., test and guidance images) are used as raw input. The resulting output is a one-dimension vector that represents the gaze direction. Likewise, the loss function used in DRNet is based on Equation (Equation 5), where α = β = 0.75. While, the loss function based on Equation (Equation 3) used in the two-stream model, where α = 0.75.

Again, we trained the two models using the two public datasets (i.e., MpiiGaze and Eyediap). In the validation step, the noise image was set as the guidance image.

Table 3 reports the performance metrics (*angular-error* and the absolute distance) using the two−stream model and DRNet, respectively. Additionally, we computed the absolute distance of difference *angular-error* for each person between the normal and noisy image. When the distance is larger, the influence of the noise image is greater. It was observed that the average *angular-error* and the absolute distance of the DRNet architecture were observed to be lower than the two−stream model using the MpiiGaze (i.e., normal image: two−stream model versus (vs) DRNet = 6.18 vs. 5.98; noisy image: two−stream model vs. DRNet = 6.39 vs. 5.99; distance (two−stream model vs. DRNet) = 0.34−0.16) and Eyediap (i.e., normal image: two−stream model vs. DRNet = 7.07 vs. 6.71; noisy image: two−stream model vs. DRNet = 7.58 vs. 6.96; distance (two−stream model vs. DRNet) = 0.73−0.41) datasets. Figure 10 shows the distance metrics using the box plot function. From the results, we conclude that the DRNet architecture provides higher performance, as shown by a lower influence of the noise image compared to the two−stream model.

### 4.3. Assessing the Impact of the Loss Functions

We conducted an experiment based on the Mnist network [17] using the loss function of LB (Equation (Equation 3)) with the MpiiGaze and Eyediap datasets. Specifically, the Mnist model uses the original loss function (Equation (Equation 2)) where α = 0 and the new loss function where α=1.

Table 4 reports the *angular-error* of the Mnist model. From the results, it was found that the MNIST model achieved the best performance with 7.27 in Eyediap and 6.07 in MpiiGaze when α in the range of [0.75,1]. We have also observed that the loss function LB provides much more optimized performance metrics.

We also studied the impact of α and β in DRNet architecture. The loss function used in DRNet with LA and LB is shown in Equation (Equation 5), Equation (Equation 4) and Equation (Equation 3), respectively. We set β and α to 0.25, 0.5, 0.75, 1. Table 5 reports the *angular-error* of DRNet in function of β and α. We found the best *angular-error* of 5.88 and 6.71 achieved when α = 0.75 and β in the range of [0.75, 1] using the MpiiGaze and Eyediap datasets, respectively. Figure 11 illustrates the surface of *angular-error* as a function with β and α. As a trade-off, we set the hyperparameters to 0.75 for both α and β.

### 4.4. Ablation Study of DRNet

We studied the impact of AD, SC and DIFF modules in the proposed DRNet architecture.

To do this, we replaced AD with a new module called DRNet_NoAD and used a parameter γ to combine DIFF and SC outputs. We have formulated the new module using Equation (Equation 7) as follows:(7)$$g_{DRNet\_NoAD}=\gamma *g_{sc}+\left(1-\gamma \right)*g_{diff}.$$
where γ is a parameter to combine $g_{sc}$ and $g_{diff}$, $g_{sc}$ and $g_{diff}$ is the output of SC and DIFF modules, $g_{DRNet\_NoAD}$ is the DRNet_NoAD output.

DRNet_NoAD used the loss function (i.e., Equation (Equation 5)) with α and β values of 0.75. Table 6, reports the *angular-error* of DRNet_NoAD. It is noted that the value of γ is automatically learned. The results showed that the performance of DRNet in terms of *angular-error* of 5.98 (MpiiGaze) and 6.71 (Eyediap) outperforms DRNet_NoAD with 6.05 (MpiiGaze) and 7.16 (Eyediap). Therefore, the AD module has demonstrated a feasible impact on the DRNet model.

A similar scenario was also considered for the SC module which was replaced by a new model, DRNet_NoSC.

We trained and tested DRNet_NoSC with the left, right, and entire eyes. Here, it is noted that the Eyediap consists of the entire left images due to the preprocessing step. Using MpiiGaze/Eyediap datasets, DRNet achieves a better performance in terms of *angular-error* of 5.98/6.71 compared to DRNet_NoSC with 6.97/7.72 and Mnist model with 6.27/7.6. The result can be shown in Table 7.

We also studied the case of replacing the DIFF module by DRNet_NoDIFF. The DRNet_NoDIFF represents the gaze direction by summing AD and SC outputs. The results in Table 7 have shown that the DRNet architecture yields better performance in terms of *angular-error* achieving 5.98/6.71 when compared to DRNet_NoDIFF with 6.07/6.97.

It is worth noting that Diff-Nn [25] achieves the lowest performance. This is due to inference related to the selection of images. This suggests that the use of a residual structure such the proposed DRNet based auxiliary information is an attractive solution. Furthermore, directly predicting difference information is not a good choice in a common environment.

## 5. Conclusions

This paper presents a novel appearance-based method (DRNet) architecture that uses the shortcut connection to combine the original gaze direction and the difference information. A new loss function is proposed to evaluate the loss in 3D space. DRNet outperforms the state-of-the-art in robustness to a noisy data set. The experimental results demonstrate that DRNet can obtain the lowest *angular-error* in MpiiGaze and Eyediap datasets by using eye features only, compared with the state-of-the-art gaze estimation methods. This paper provides a feasible solution to address the gaze calibration problem and enhance the robustness of noise images. In future work, we will consider more factors in our DRNet model to improve the performance metrics, in particular, when we use facial and eye features.

## Figures and Tables

**Figure 1 sensors-22-05462-f001:**
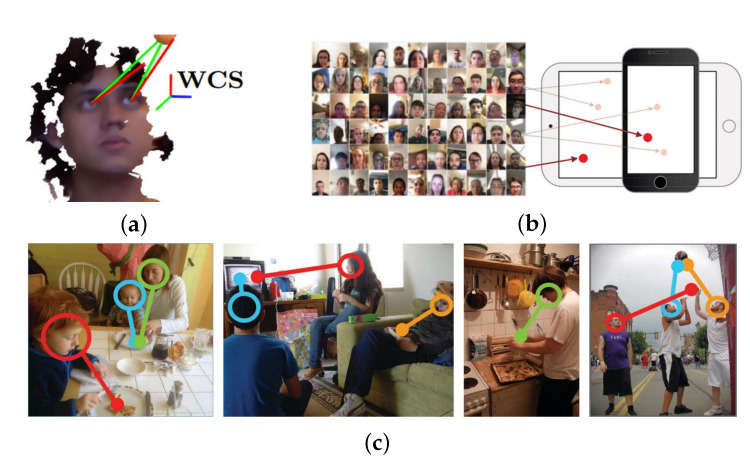
Examples of common tasks of gaze estimation: (**a**) three-dimensional (3D)-based estimation [9], (**b**) target estimation [10,11] and (**c**) tracking [12].

**Figure 2 sensors-22-05462-f002:**
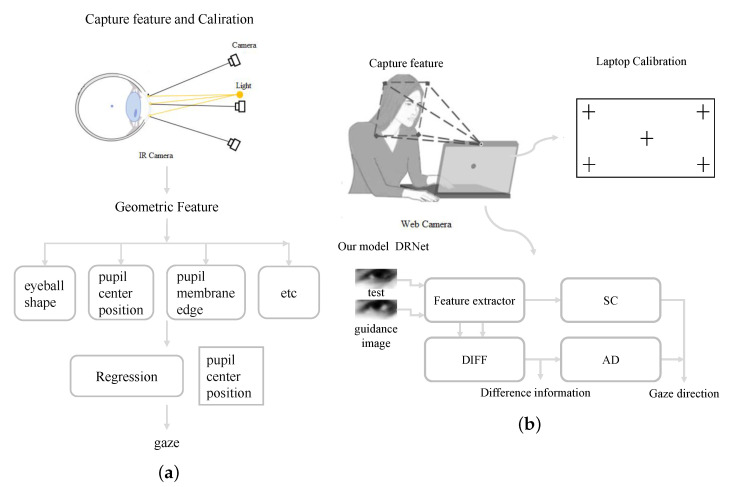
Example of two 3D gaze estimation techniques, (**a**) model-based and (**b**) appearance-based methods.

**Figure 3 sensors-22-05462-f003:**
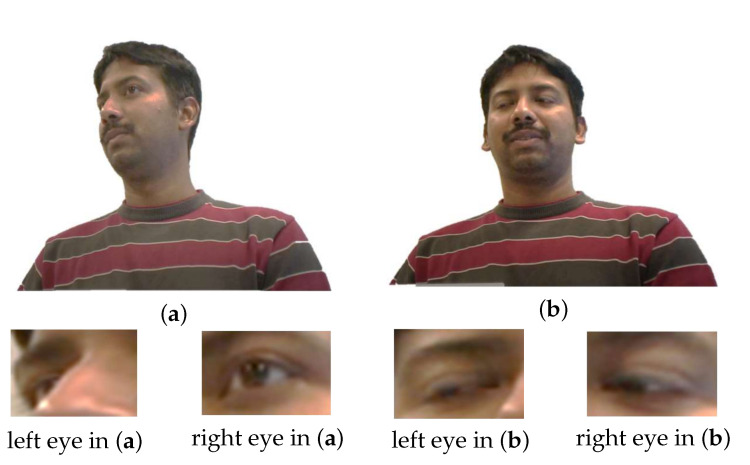
Example of noise images in Eyediap [20]. (**a**,**b**) are two example frames of the dataset. (**Top**) Original RGB frames. (**Bottom**) Left- and right-eye images are captured from the original RGB frames.

**Figure 4 sensors-22-05462-f004:**
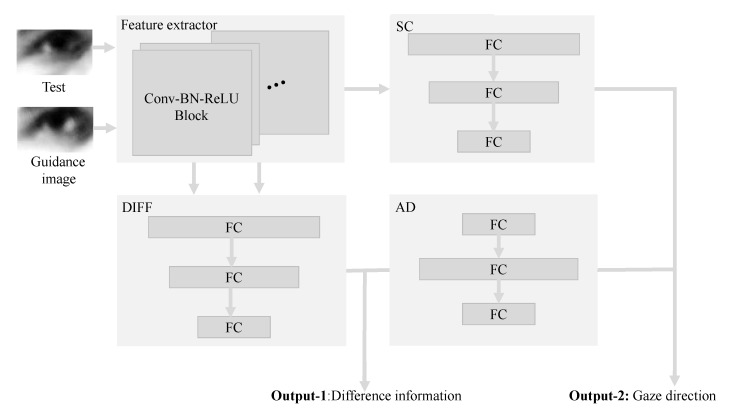
The proposed DRNet pipeline. Test and guidance images used as input to the DRNet model. DRNet provides the difference information and the gaze direction, which are described by three-dimensional vectors.

**Figure 5 sensors-22-05462-f005:**
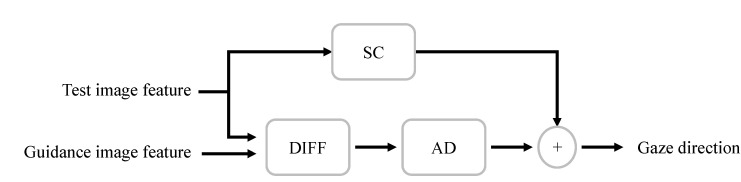
DRNet residual structure. The guidance image and the test image features are extracted from the raw inputs. The sum of SC and AD outputs provides the gaze direction.

**Figure 6 sensors-22-05462-f006:**
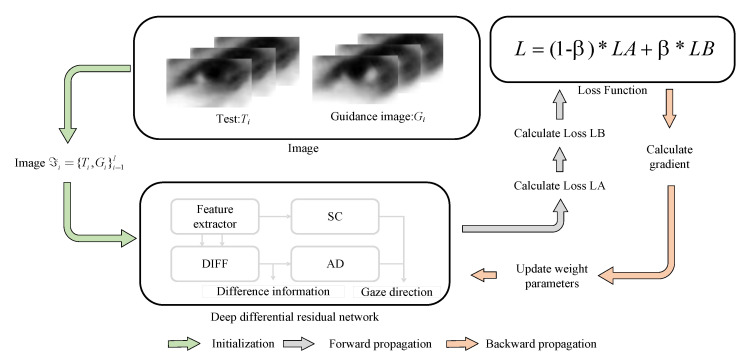
Flow chart of the training stage.

**Figure 7 sensors-22-05462-f007:**
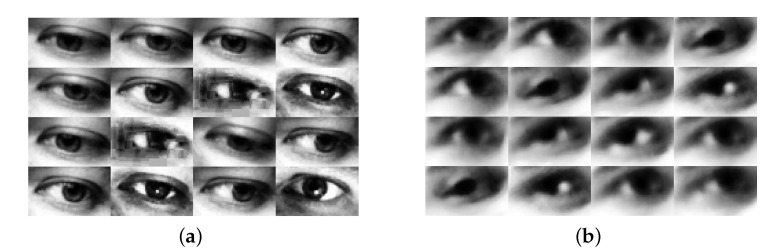
Sample images from MpiiGaze (**a**) and Eyediap (**b**).

**Figure 8 sensors-22-05462-f008:**
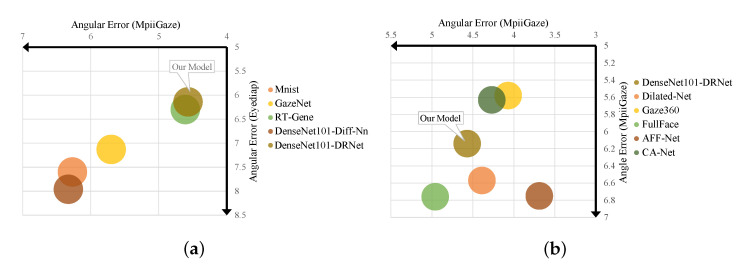
Performance (*angular-error*) of methods that use (**a**) eye features or (**b**) combined eye and facial features. The horizontal axis records the error in the MpiiGaze dataset and the vertical axis records the error in the Eyediap dataset. The model closest to the upper right corner represents better performance.

**Figure 9 sensors-22-05462-f009:**
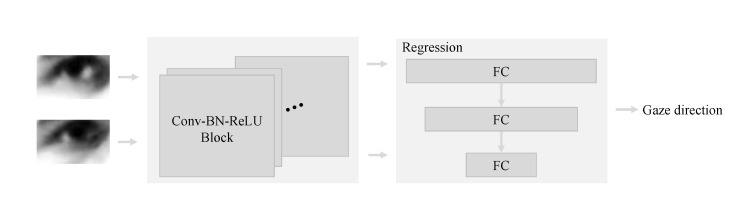
Example of a two-stream model. The feature extractor module is based on convolution layers. Regression module is represented by fully connected layers.

**Figure 10 sensors-22-05462-f010:**
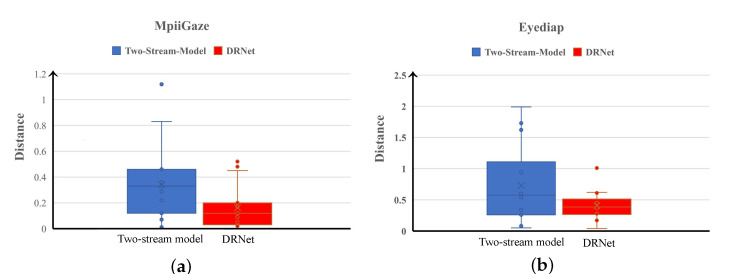
Box plots of the absolute distance of *angular-error* between the normal and noisy image using the two-stream model and DRNet in MpiiGaze (**a**) and Eyediap (**b**).

**Figure 11 sensors-22-05462-f011:**
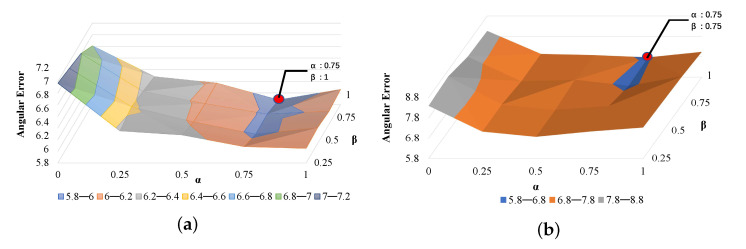
Surface plots the *angular-error* in function with α and β using MpiiGaze (**a**) and Eyediap (**b**). The red dot represents the best performance.

**Table 1 sensors-22-05462-t001:** Performance summary (*angular-error*) and comparison with recent works using eye features.

Method	MpiiGaze	Eyediap
Mnist [18]	6.27	7.6
GazeNet [17]	5.7	7.13
RT-Gene [27]	4.61	6.3
DenseNet101-Diff-Nn [25]	6.33	7.96
DenseNet101-DRNet (ours)	**4.57**	**6.14**

**Table 2 sensors-22-05462-t002:** Performance summary (*angular-error*) and comparison with recent works using eye and facial features.

Method	MpiiGaze	Eyediap
Dilated-Net [31]	4.39	6.57
Gaze360 [39]	4.07	**5.58**
FullFace [29]	4.96	6.76
AFF-Net [34]	**3.69**	6.75
CA-Net [30]	4.27	5.63

**Table 3 sensors-22-05462-t003:** Summary of *angular-error* and absolute distance for robustness evaluation in the two-stream and DRNet model.

Two Stream Model versus DRNet			
**MpiiGaze**	No_Invalid_Image	Fixed_Invalid_Image	Distance
$P_{00}$	4.72-4.46	5.09-4.48	0.37-0.02
$p_{01}$	5.98-5.99	6.11-6.02	0.13-0.03
$p_{02}$	5.29-5.02	5.41-4.85	0.12-0.17
$p_{03}$	6.65-6.61	6.43-6.41	0.22-0.20
$p_{04}$	6.78-6.70	6.79-6.18	0.01-0.52
$p_{05}$	6.20-6.27	7.32-6.39	1.12-0.12
$p_{06}$	6.15-5.94	6.61-5.94	0.46-0.00
$p_{07}$	7.44-7.19	7.11-7.07	0.33-0.12
$p_{08}$	6.51-6.46	6.84-6.41	0.33-0.05
$p_{09}$	7.98-7.07	7.91-7.09	0.07-0.02
$p_{10}$	5.41-5.38	5.88-5.33	0.47-0.05
$p_{11}$	5.26-4.88	5.26-5.36	0.00-0.48
$p_{12}$	5.87-5.33	6.70-5.78	0.83-0.45
$p_{13}$	6.39-6.11	6.68-6.20	0.29-0.09
$p_{14}$	6.01-6.22	5.65-6.34	0.36-0.12
Average	6.18-5.98	6.39-5.99	0.34-**0.16**
**Eyediap**	No_Invalid_Image	Fixed_Invalid_Image	Distance
$p_{1}$	7.35-6.86	7.27-7.14	0.08-0.28
$p_{2}$	7.43-7.33	6.87-7.66	0.56-0.33
$p_{3}$	5.78-6.01	6.41-6.05	0.63-0.04
$p_{4}$	7.66-5.29	7.92-5.58	0.26-0.29
$p_{5}$	8.08-6.06	8.67-7.07	0.59-1.01
$p_{6}$	7.14-5.84	7.40-6.21	0.26-0.37
$p_{7}$	7.64-6.96	9.63-7.58	1.99-0.62
$p_{8}$	8.23-5.44	9.17-5.61	0.94-0.17
$p_{9}$	8.10-7.37	7.56-7.77	0.54-0.40
$p_{10}$	7.24-7.87	8.86-8.32	1.62-0.45
$p_{11}$	6.46-6.93	6.12-7.54	0.34-0.61
$p_{14}$	5.35-7.78	5.30-7.56	0.05-0.22
$p_{15}$	6.11-7.26	7.84-6.78	1.73-0.48
$p_{16}$	6.46-7.00	7.06-6.58	0.60-0.42
Average	7.07-6.71	7.58-6.96	0.73-**0.41**

No_Invalid_Image and Fixed_Invalid_Image represent the normal and noisy image, respectively. Distance represents the absolute value of difference *angular-error* for each person between No_Invalid_Image and Fixed_Invalid_Image. (-): versus.

**Table 4 sensors-22-05462-t004:** The performance of Mnist [5] with different α.

α	Eyediap	MpiiGaze
1	7.31	6.07
0.75	7.27	6.12
0.5	7.38	6.25
0.25	7.59	6.53
0	7.6	6.3

**Table 5 sensors-22-05462-t005:** The performance of DRNet in MpiiGaze/Eyediap with different α and β.

	β	0.25	0.5	0.75	1
α					
0		7.17/8.4	7.09/8,5	7.01/8.14	6.81/8.07
0.25		6.35/7.13	6.43/7.13	6.36/6.93	6.28/6.94
0.5		6.28/6.87	6.17/6.87	6.15/7.01	6.17/6.88
0.75		6.08/7.13	5.96/7.06	5.97/**6.71**	**5.88**/6.77
1		6.05/7.33	6.07/7.26	6.02/7.18	6.06/7.04

**Table 6 sensors-22-05462-t006:** The performance of DRNet_NoAD.

Dataset	Angular−Error	γ
MpiiGaze	6.05	0.89
Eyediap	7.16	0.88

**Table 7 sensors-22-05462-t007:** The performance of Diff-Nn in common environment.

Method	MpiiGaze (L/R/All)	Eyediap (L/R/All)
DRNet	6.15/6.29/5.98	6.71/-/-
Diff-Nn [25]	10.73/10.92/10.83	11.82/-/-
DRNet_NoSC	6.59/6.32/6.97	7.72/-/-
DRNet_NoAD	6.11/6.25/6.05	7.16/-/-
DRNet_NoDIFF	6.22/6.32/6.07	6.97/-/-

## Data Availability

MpiiGaze openly available in a public repository. The MpiiGaze that support the findings of this study are openly available at https://www.mpi-inf.mpg.de/departments/computer-vision-and-machine-learning/research/gaze-based-human-computer-interaction/appearance-based-gaze-estimation-in-the-wild. Eyediap available on request from the authors. The data that support the findings of this study are available from the corresponding author (Y.L.) upon reasonable request.

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
