# Peer review of "Gaze Estimation Approach Using Deep Differential Residual Network"

_sensors, 2022, doi:10.3390/s22145462_

Round 1

Reviewer 1 Report

Authors present a new method to gaze estimation. They propose a new loss fonction and the  DRNet model to improve improve the robustness-to-noise image in  eye images.

The work is soundness and interesting but in some parts the descrption of the method is quite confuse :

For exemple, when the authors describe Fig. 4 in the text appears terms as DIFF, AD and SC but in the Figure the terms are missing. 

Explain better the results of Table 2. Authos said that the results are competitive but they don´t explain why their score are not more competitive.

Conclusion are not very detailed.

Check the english: syntax and style.

Reviewer 2 Report

This study is devoted to the development of methods for estimating the gaze of a person. The direction of gaze is one of the important characteristics in the formation of human behavior patterns. The study relevance is due to the ability to estimate the focus of a person's attention to determine his interest, provide targeted support in information systems, and improve interaction interfaces in a pair of man-machine. The proposed improvement of methods to analyze the direction of a person’s gaze occurs by reducing the noise in the image of the user's eyes, which is achieved by modifying the applied differential residual model of deep learning. The paper provides a comprehensive analysis of works related to the research topic. The main trends in the development of the problem area are identified. Solutions review  obtained earlier is given and the remaining problems are highlighted. All the results stated in the article are confirmed and can be repeated and interpreted in the applied problems. The loss function replacement essence for the target machine learning model in the human gaze analysis is fully disclosed. The paper is well structured. The research materials presentation is consistently and easily explained, all the necessary graphic materials and calculations are given.

There are minor comments that do not affect the semantic content of the work :

1. More clearly state the abstract, partially adding the results obtained.

2. Reveal the conclusion. For example, to disclose in more detail the results obtained and their interpretability in further research.

The new scientific results obtained in the course of the study may be useful for solving specific applied problems. The development of computer vision systems in the modern world plays an enormous role in solving social, economic, and industrial problems. This study meets the requirements for publication in a peer-reviewed scientific publication.

Reviewer 3 Report

In this paper, the authors have proposed gaze estimation using the deep differential residual network. Overall, the paper is well written but requires some major improvements which are mentioned below.

1) In the abstract, please include the details of the dataset that you used and also mention the quantitative results with the proposed method.

2) In the related work, also discuss the studies related to gaze estimation in outdoor environments such as "gaze detection systems for automobile drivers" and "drivers emotions classification" using gaze. 

3) Please verify your results on the open Columbia gaze dataset (CAVE-DB).

Round 2

Reviewer 3 Report

Most of my comments are addressed. I recommend acceptance of this article in its current form.